# Exploiting Tree Structure for Credit Assignment in RL Training of LLMs

## Abstract

Reinforcement learning improves LLM reasoning, yet sparse delayed reward over long sequences makes token-level credit assignment the key bottleneck. We study the verifiable-reward setting, where the final answer is checkable and multiple responses can be drawn per prompt. Reasoning tasks in math and medical QA align with this setup, where only a few decision tokens significantly impact the outcome. PPO offers token-level advantages with a learned value model, but it is complex to train both the actor and critic models simultaneously, and it is not easily generalizable, as the token-level values from the critic model can make training prone to overfitting. GRPO is critic-free and supports verifiable rewards, but spreads a single sequence-level return across tokens and ignores branching. We introduce **Prefix-to-Tree (P2T)**, a simple procedure that converts a group of responses into a prefix tree and computes *nonparametric* prefix values $V(s)$ by aggregating descendant outcomes. Built on P2T, we propose **TEMPO** (*Tree-Estimated Mean Prefix Value for Policy Optimization*), a critic-free algorithm that augments the group-relative outcome signal of GRPO with *branch-gated* temporal-difference corrections derived from the tree. At non-branch tokens, the temporal-difference (TD) term is zero, so TEMPO reduces to GRPO; at branching tokens, it supplies precise token-level credit without a learned value network or extra judges/teachers. On Qwen3-1.7B/4B, TEMPO outperforms PPO and GRPO on in-distribution (MATH, MedQA) and out-of-distribution (GSM-HARD, AMC23, MedMCQA, MMLU-Medical) benchmarks, and reaches higher validation accuracy with less wall-clock time. [1]

"In chess, a tempo is a move saved at a fork; in learning, credit should fall on that move."

## 1 Introduction

Reinforcement learning (RL) (Sutton et al., 1998) is an effective way to strengthen the reasoning of large language models (LLMs) (Zhang et al., 2025). In LLM settings, rewards are sparse and delayed and sequences are long (Jaech et al., 2024; Guo et al., 2025), so the key challenge is **credit assignment**: give the outcome reward to the few tokens that really change the solution. We study the *verifiable-reward* setting, where the final answer for a prompt is checkable and we can draw multiple responses for the same prompt. This is common in long "thinking" or chain-of-thought (CoT) tasks such as mathematics and medical QA, where most steps are low-impact and only a small set of *decision tokens* (e.g., strategy choice, formula selection, diagnostic commitment) moves the outcome. Aggregating multiple responses naturally induces an *implicit prefix tree*: internal nodes are shared prefixes and branch nodes mark decision points with multiple plausible continuations. A good learning rule should use the branching structure across responses and focus credit on those decision points.

**Proximal Policy Optimization(PPO)** gives token-level advantages with a learned value and generalized advantage estimation (GAE), which mixes Monte Carlo (MC) returns with temporal-difference (TD) bootstrapping (Schulman et al., 2015; 2017). However, jointly training the actor and critic is complex and often fails to generalize, as critic-derived token-level values can induce overfitting Wang et al. (2025b); Chaudhari et al. (2024). **Group Relative Policy Optimization(GRPO)**

---
[1]Our code can be accessed at: `https://anonymous.4open.science/r/tempo-0AF2`

removes the critic and uses group-relative baselines over responses to the same prompt (Shao et al., 2024; Yu et al., 2025). It is simple and fits verifiable rewards. However, it spreads a single sequence-level signal across all tokens and overlooks mid-trajectory decisions. As a result, token-level credit is weak when reasoning branches. Recent "key-token ideas" (Wang et al., 2025a) move toward finer signals by focusing gradient updates on high-entropy tokens. This approach benefits from exploiting response structure and concentrating learning on decision-heavy positions. However, while entropy-based updates can improve exploitation within known reasoning patterns, they may struggle to acquire new domain knowledge where exploration across broader patterns of the responses is necessary.

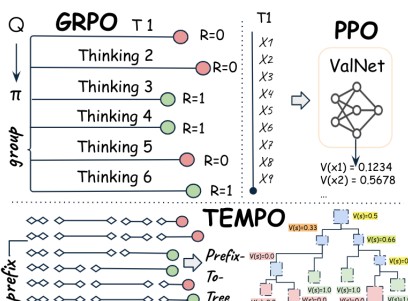

We present **Prefix-to-Tree (P2T)** as a simple procedure that converts a group of responses into a prefix tree and computes *nonparametric* prefix values $V(s)$ by averaging descendant returns. Building on P2T, we introduce **TEMPO** (*Tree-Estimated Mean Prefix Value for Policy Optimization*), a critic-free policy optimization method that restores token-level credit only where it matters. For each prompt, sampled responses form paths in the implicit prefix tree. TEMPO augments the group-relative outcome signal of GRPO with *branch-gated* temporal-difference (TD) corrections derived from the tree: at non-branching tokens $V(s_{t+1}) = V(s_t)$ and the TD error is zero, so the update reduces to GRPO, while at branching tokens it supplies precise token-level credit. TEMPO maintains the GRPO training loop and cost. It does not train a value model or add a process reward model or a judge, it also does not require a teacher or a new sampler.

Figure 1: **Comparison of credit assignment for RL training with verifiable rewards.** GRPO: all tokens in each sampled answer share one sequence-level return; branching is ignored so credit spreads evenly. PPO: a learned value network estimates $V(s_t)$ and provides token-level advantages via GAE, but requires a critic and higher compute. TEMPO: convert the answer group for one prompt into a prefix tree and compute *nonparametric* prefix values $V(s)$ by averaging descendant outcomes; use *branch-gated* TD corrections to assign credit at branches.

**Empirical scope and applicability**. Across *Qwen3-1.7B* and *Qwen3-4B*, TEMPO consistently attains higher accuracy than PPO (Schulman et al., 2017), GRPO (Shao et al., 2024), and HEPO (Wang et al., 2025a) on both in-distribution (MATH, MedQA) and out-of-distribution (GSM-HARD, AMC23, MedMCQA, MMLU-Medical) evaluations, while reaching strong validation performance in less wall-clock time. Validation curves indicate that, on math reasoning, approaches that emphasize token-level structure, such as HEPO (Wang et al., 2025a) (e.g., focusing updates on high-entropy decision tokens) already enjoy an advantage, suggesting the RL phase mainly reinforces reasoning patterns learned during pretraining and SFT. Yet, TEMPO goes further by injecting tree-gated TD credit at the exact branching points. On medical reasoning, where domain knowledge must be newly acquired, methods that rely on group-relative exploration such as GRPO (Shao et al., 2024) generalize better than purely exploitation-oriented updates; TEMPO combines this robust group baseline with branch-aware TD from P2T's nonparametric prefix values, improving both convergence speed and final accuracy. In practice, TEMPO is most beneficial when rewards are verifiable and prompts yield meaningful branching, delivering precise token-level credit without a value network or auxiliary judges, and serving as a drop-in, efficiency-preserving upgrade to GRPO-style training. Overall, our key **contributions** include:

1. We introduce **Prefix-to-Tree (P2T)**, a simple procedure that converts each prompt's group of responses into a prefix tree and derives *nonparametric* prefix values $V(s_t)$ by aggregating descendant outcomes.

2. Building on P2T, we propose **TEMPO**, a drop-in, GRPO-compatible algorithm that augments the group-normalized outcome signal with *branch-gated* TD corrections, providing precise token-level credit at decision points while retaining GRPO-like compute and simplicity.

3. On *Qwen3-1.7B/4B*, TEMPO improves convergence speed and final accuracy over other baselines on in-distribution (MATH, MedQA) and out-of-distribution (GSM-HARD, AMC23, MedMCQA, MMLU-Medical) benchmarks under the same hardware budget.

## 2 RELATED WORK

Credit assignment is central in post-training for reasoning LLMs. RLHF brought PPO with a learned value (critic) and GAE to reduce variance (Ouyang et al., 2022; Schulman et al., 2017; 2015). This improved alignment, however, comes at the cost of critic training, which adds complexity and tuning, and value prediction is brittle on long chains. To avoid a critic, several lines move towards value-free or RL-free updates that treat the entire response as a single action. DPO optimizes pairwise preferences in an offline bandit view (Rafailov et al., 2023). Rejection-sampling methods, such as RestEM, fine-tune only on full high-reward responses (Singh et al., 2023). RLOO, GRPO, and DAPO compute group-normalized sequence advantages over multiple samples of the same prompt, thereby removing the value network (Ahmadian et al., 2024; Shao et al., 2024; Yu et al., 2025). These methods are simple and stable, but their feedback is sequence-level and credits all tokens equally, which weakens token-level credit in long reasoning. VinePPO instead replaces the critic with Monte Carlo estimates obtained by re-sampling continuations from each text prefix, yielding accurate prefix values in language settings (Kazemnejad et al., 2024). However, it requires fresh rollouts at many branch nodes and raising sampling cost when trees are wide or branch early and still uses path-wise PPO advantages without group-normalized baselines.

Several contemporaneous works exploit the *tree structure* of rollouts to densify credit assignment and/or cut sampling cost. TreePO Li et al. (2025) reframes on-policy rollouts as a tree search with segmented decoding and heuristic branching/fallback, amortizing shared prefixes (KV caching) and introducing a tree-based segment-level advantage estimator; this improves stability and reduces sampling compute while maintaining or improving accuracy. TREERPO Yang et al. (2025b) extends GRPO by performing explicit tree sampling and forming step-level sibling groups to estimate expected rewards per step, yielding dense process signals and reporting consistent gains over GRPO with shorter responses. TreeRL Hou et al. (2025) integrates an entropy-guided sampler (EPTree) that branches at uncertain tokens, then back-propagates leaf rewards to provide global and local (step) advantages thereby eliminating a separate process reward model. Tree-OPO Huang et al. (2025) leverages *off-policy* teacher MCTS to build prefix trees and proposes staged, prefix-conditioned advantage estimation to stabilize GRPO-style updates. Unlike methods that require dedicated tree samplers (TreeRPO/TreeRL) or off-policy teacher trees (Tree-OPO), TEMPO operates in the standard GRPO setting and treats the *implicit* prefix tree formed by a group of responses as a nonparametric value baseline: it computes $V(s_t)$ from all completions sharing the prefix $s_t$ and adds a token-level TD correction to the group-relative (Monte Carlo) signal. This yields branch-aware advantages without a learned value network, extra reward/process models, or special search procedures, while remaining fully on-policy and drop-in compatible with GRPO training loops.

## 3 PRELIMINARIES

We begin by reviewing the advantage estimation used in Proximal Policy Optimization and Group Relative Policy Optimization.

**PPO.** PPO Schulman et al. (2017) is a policy gradient method that stabilizes updates via a clipped objective. A key component is the estimation of the advantage function $A_t$, which measures how much better an action $a_t$ is compared to the average action at state $s_t$. PPO commonly employs *generalized advantage estimation* (GAE) Schulman et al. (2017), defined as

$$\hat{A}_t^{\mathrm{GAE}(\gamma,\lambda)} = \sum_{l=0}^{T-t-1} (\gamma\lambda)^l \, \delta_{t+l}, \qquad \delta_t = r_t + \gamma V(s_{t+1}) - V(s_t).$$

In the original formulation, $\gamma$ serves as a discount factor that reduces the weight of delayed rewards and helps stabilize infinite-horizon settings. However, in the context of large language model (LLM) training, it is common to set $\gamma = 1.0$ so that long completions are not penalized relative to short ones. With this setting, the GAE formula simplifies to

$$\hat{A}_t^{\mathrm{GAE}(\lambda)} = \sum_{l=0}^{T-t-1} \lambda^l \, \delta_{t+l}, \qquad \delta_t = r_t + V(s_{t+1}) - V(s_t).$$

The bias–variance tradeoff is then controlled solely by the parameter $\lambda$.

**Special cases.**

- $\lambda = 0$ *(TD(0)).*
$$\hat{A}_t^{\lambda=0} = \delta_t = r_t + V(s_{t+1}) - V(s_t),$$
  the one-step temporal-difference error (lowest variance, highest bias).

- $\lambda = 1$ *(Monte Carlo).*
$$\hat{A}_t^{\lambda=1} = \sum_{l=0}^{T-t-1} r_{t+l} + V(s_T) - V(s_t).$$

  If $s_T$ is terminal so $V(s_T) = 0$, then
$$\hat{A}_t^{\lambda=1} = \Big( \sum_{l=0}^{T-t-1} r_{t+l} \Big) - V(s_t),$$

  i.e., the full Monte Carlo return minus the baseline (unbiased, higher variance).

**GRPO.** GRPO Shao et al. (2024) was designed for reinforcement learning with verifiable feedback. Formally, for each question $q$, a group of $G$ responses $\{o_1, \ldots, o_G\}$ is sampled from the old policy $\pi_{\theta_{\text{old}}}$, and a reward model assigns scores $r = \{r_1, \ldots, r_G\}$. These rewards are then normalized by subtracting the group mean and dividing by the group standard deviation. Under outcome supervision, the normalized reward $\tilde{r}_i = \frac{r_i - \text{mean}(r)}{\text{std}(r)}$ is applied uniformly to all tokens of output $o_i$, so that
$$\hat{A}_{i,t} = \tilde{r}_i = \frac{r_i - \text{mean}(r)}{\text{std}(r)}, \quad \forall t \in o_i.$$

Under process supervision, the same normalization is applied at the step level, and the normalized step rewards are distributed to the corresponding tokens. GRPO thus avoids training a separate value model and provides efficient group-relative baselines, but it relies purely on Monte Carlo outcomes and discards token-level temporal structure. This setup highlights the gap: PPO provides token-level advantages but requires a value model, while GRPO is model-free but trajectory-level only. Our method, TEMPO, combines the strengths of both.

## 4 METHODOLOGY

### 4.1 VALUE ESTIMATION FROM PREFIX TREE

Figure 2 illustrates how TEMPO derives value estimates directly from the tree structure formed by a group of sampled responses. Each path in the tree corresponds to a response generated by the policy, and each node represents a token prefix $s_t$ up to time $t$. The tree branches whenever different responses diverge at a given token. Terminal nodes are assigned rewards $r \in \{0, 1\}$ based on verifiable correctness (e.g., whether the final answer matches the ground truth).

Instead of training a separate value model as in PPO, TEMPO computes $V(s_t)$ directly from the group of trajectories. For a given prefix $s_t$, the value is estimated as the average normalized reward of all descendant completions that share this prefix:

$$V(s_t) = \frac{1}{|D(s_t)|} \sum_{j \in D(s_t)} r_j,$$

where $D(s_t)$ is the set of responses passing through $s_t$, and $r_j$ is the outcome reward. This provides a *value function* without introducing an additional learned critic.

In the example shown in Figure 2, some prefixes lead to correct answers ($r = 1$) while others lead to incorrect ones ($r = 0$). TEMPO propagates these signals upward by averaging over the subtree, yielding intermediate values (e.g., $V(s_t) = 0.5$ when half of the descendant completions are correct). As a result, branch nodes obtain informative value estimates that reflect the quality of their

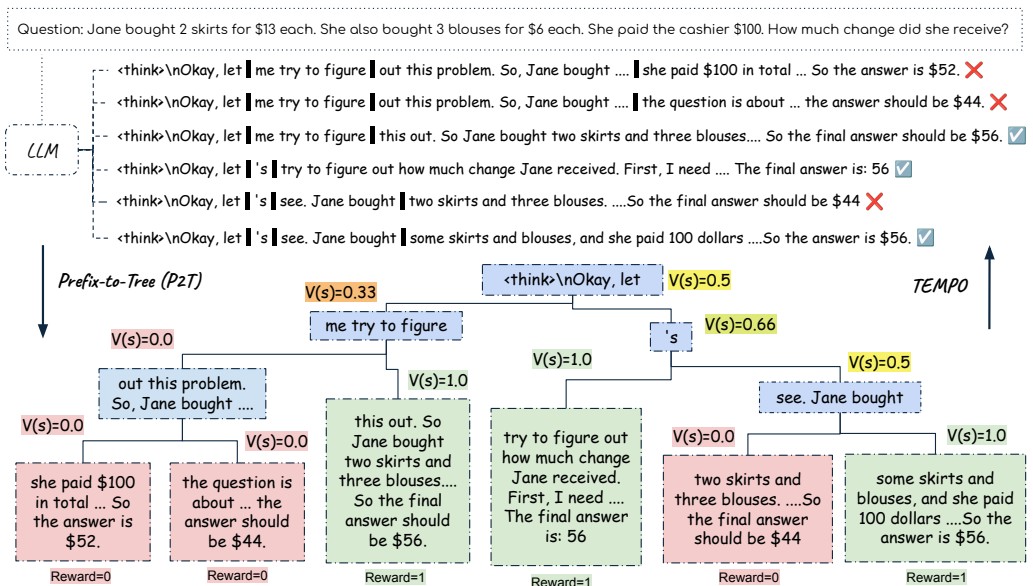

Figure 2: Overview of prefix tree value estimation in TEMPO. Each node corresponds to a token prefix $s$, with $V(s)$ estimated by averaging over the outcomes of all descendant completions. Green leaves denote correct responses ($r = 1$), red leaves denote incorrect ones ($r = 0$). Intermediate nodes inherit averaged values (e.g., $V(s) = 0.5$), providing informative signals at branching points. '

continuations, while non-branch nodes naturally inherit consistent values from their unique continuation. This design ensures that tokens along successful reasoning paths (green leaves) contribute positively to the estimated value of their prefixes, tokens along failed reasoning paths (red leaves) reduce the value of their prefixes and branching points receive *discriminative signals*, as the value function captures how sibling continuations differ in correctness. By computing $V(s_t)$ directly from the tree, TEMPO provides token-level evaluative feedback while remaining model-free, combining the efficiency of GRPO with the structured credit assignment of PPO.

## 4.2 BRANCH-AWARE ADVANTAGE ESTIMATION

Having defined the prefix tree value function $V(s_t)$, we now describe how TEMPO constructs advantages by combining *response-level Monte Carlo signals* from GRPO with *token-level temporal-difference corrections* derived from the tree.

**Response-level (MC) signal.** GRPO provides outcome-level supervision by normalizing the rewards across a group of $G$ responses. For outcome supervision, each response $o_i$ receives a normalized reward

$$\tilde{r}_i = \frac{r_i - \text{mean}(r)}{\text{std}(r)},$$

and assigns it uniformly to all tokens of the trajectory. This yields a pure Monte Carlo signal: every token in a response inherits the same scalar advantage $\tilde{r}_i$. While efficient, this discards the structure of reasoning trajectories.

**Token-level (TD) correction.** TEMPO augments this outcome-level signal with a token-level TD term based on branch-aware values. For token $t$ in trajectory $i$, with state prefix $s_t$ and successor $s_{t+1}$, we define the TD error as

$$\delta_{i,t} = V(s_{t+1}) - V(s_t).$$

This term captures how much the estimated value changes when extending from prefix $s_t$ to $s_{t+1}$. Importantly, $\delta_{i,t}$ is only nonzero at branching points, since non-branch tokens have identical descendant outcomes and thus $V(s_{t+1}) = V(s_t)$.

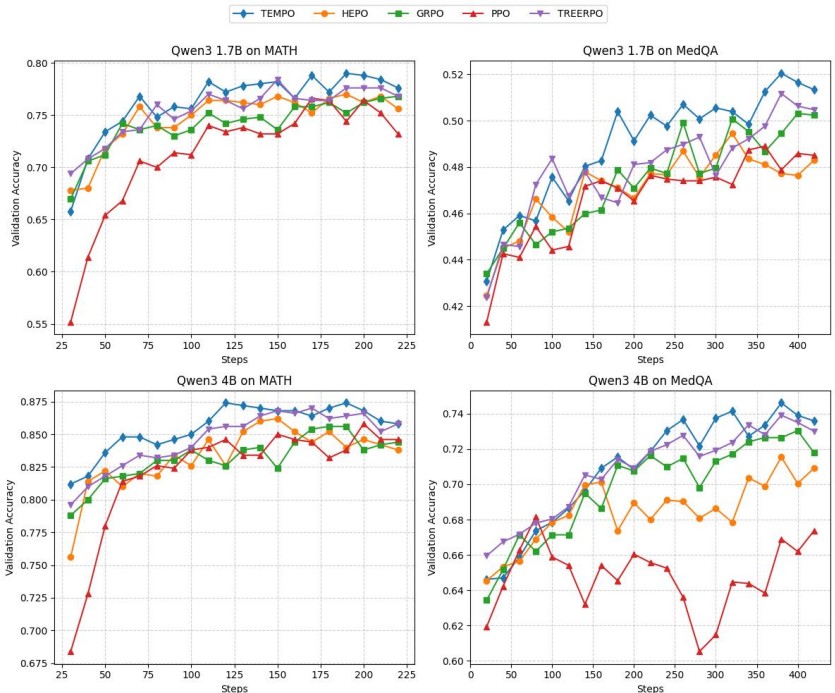

Figure 3: Validation accuracy of MATH and MedQA for Qwen3-1.7B and Qwen3-4B. We compare TEMPO with PPO, GRPO, and HEPO. TEMPO consistently achieves higher accuracy and faster convergence across both domains and model sizes.

**Combined TEMPO advantage.** The final TEMPO advantage integrates both levels of signal:

$$\hat{A}_{i,t} = \frac{1}{\text{std}(r)}[\underbrace{r_i - \text{mean}(r)}_{\text{MC signal}} + \underbrace{V(s_{t+1}) - V(s_t)}_{\text{TD error}}]$$

The *MC signal* provides global outcome-level supervision, aligning each response relative to its group. The *TD error* provides local, branch-aware token-level feedback, highlighting where reasoning paths diverge in quality.

### 4.3 POLICY UPDATE

For the policy update, TEMPO follows the same principles as DAPO Yu et al. (2025), incorporating several practical design choices that improve stability and efficiency such as Clip-Higher (decoupled clipping), token-level policy-gradient loss (global token averaging) and remove KL divergence.

**TEMPO loss function.** Combining these practices with our proposed branch-aware advantage estimation, the loss is defined as

$$\mathcal{J}_{\text{TEMPO}}(\theta) = \mathbb{E}_{q, \{o_i\}_{i=1}^G \sim \pi_{\theta_{\text{old}}}(\cdot|q)}\left[\frac{1}{\sum_{i=1}^G |o_i|}\sum_{i=1}^G \sum_{t=1}^{|o_i|} \min\Big(r_{i,t}(\theta)\,\hat{A}_{i,t},\right.$$

$$\left.\text{clip}\big(r_{i,t}(\theta),\,1-\varepsilon_{\text{low}},\,1+\varepsilon_{\text{high}}\big)\,\hat{A}_{i,t}\Big)\right], \qquad (1)$$

## 5 EXPERIMENTAL SETUP

**Datasets and Models** We train on MATH (Hendrycks et al.) and MedQA (Jin et al., 2021) as in-distribution tasks, and evaluate on their test sets plus OOD benchmarks (GSM-HARD, AMC23,

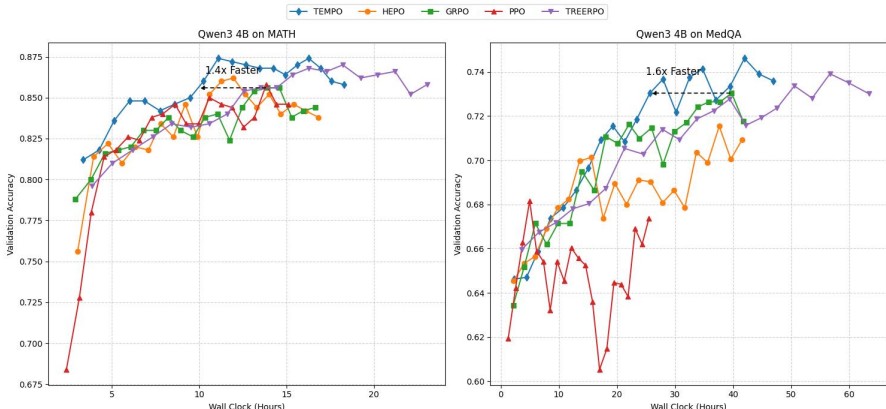

Figure 4: TEMPO converges faster and to higher accuracy than GRPO, passes GRPO's peak performance in fewer iterations and less overall time.

MedMCQA, MMLU-Medical). Experiments use Qwen3-1.7B and Qwen3-4B (Yang et al., 2025a) under identical settings(details in Appendix A.2).

**Baselines**   Our main baseline is GRPO (Shao et al., 2024), which incorporates several practical strategies from DAPO (Yu et al., 2025): removing the KL penalty, introducing a clip-higher mechanism, and applying a token-level policy gradient loss. These modifications make GRPO one of the state-of-the-art RLVF algorithms without requiring a value network. We further include TREERPO Yang et al. (2025b), which constructs sampled trees and computes step-level, group-relative rewards to provide denser credit signals without a separate reward model Yang et al. (2025b). TEMPO builds on GRPO and improves credit assignment by exploiting the tree structure of responses. We also compare against a GRPO variant that targets *high-entropy minority tokens* (Wang et al., 2025a), where gradient updates are applied only to high-entropy tokens. For clarity in experiments and figures, we denote this variant as HEPO (High Entropy Policy Optimization). Finally, we include an actor–critic baseline: PPO (Schulman et al., 2017), where the critic model is matched in size to the actor model.

## 6   RESULTS

In this section, we evaluate the effect of better Credit Assignment on task performance, efficiency, and generalization dynamics.

### 6.1   TASK PERFORMANCE

Figure 3 shows validation curves on MATH and MedQA for both Qwen3-1.7B&4B. Across settings, TEMPO has the best performance in terms of convergence speed and final accuracy. On MATH, TEMPO consistently outperforms others, followed by TREERPO, then HEPO, GRPO, and PPO. The fact that HEPO performs slightly better than GRPO and PPO suggests that focusing updates on high-entropy tokens helps exploit the reasoning structures already present in the model. Since mathematical reasoning knowledge is largely captured during pretraining and instruction tuning, the RL process primarily reinforces existing structures rather than learning new ones. Thus, token–structure–oriented methods, such as HEPO, gain an advantage compared with GRPO. On MedQA, however, the trend differs. TEMPO again delivers the best results, but GRPO surpasses HEPO, and PPO lags behind. We hypothesize that medical reasoning requires learning novel domain-specific knowledge, which emphasizes *exploration* rather than pure exploitation of existing token-level structures. Here, GRPO's group-relative normalization provides stronger signals than PPO, while HEPO's focus on high-entropy tokens is insufficient to capture new knowledge. TEMPO combines the benefits of GRPO with tree-structured TD guidance, enabling effective exploration while still leveraging structural signals, leading to the best generalization in the medical domain.

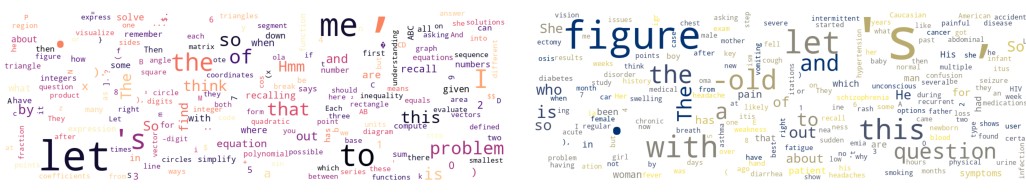

| (a) MATH (branching tokens) | (b) MedQA (branching tokens) |

Figure 6: Word clouds of *branching tokens* (multi-child nodes) collected during TEMPO training. Token sizes reflect frequency among branch points across training steps. In both domains, branching tokens align with decision points where the reasoning path can fork.

| Model | Math | | | Medical | | |
|---|---|---|---|---|---|---|
| | **MATH** | **GSM-HARD** | **AMC23** | **MedQA** | **MedMCQA** | **MMLU-Medical** |
| Qwen3-1.7B | 68.5 | 46.85 | 57.5 | 46.11 | 43.17 | 57.85 |
| + PPO | 81.6 | 53.37 | 67.5 | 52.94 | 48.05 | 70.16 |
| + GRPO | 82.4 | 53.15 | 72.5 | 56.24 | 49.56 | 71.53 |
| + HEPO | 81.7 | 52.09 | 62.5 | 54.28 | 48.98 | 71.35 |
| + TREERPO | 84.6 | 54.82 | 72.5 | 57.42 | 50.23 | 72.08 |
| + TEMPO | **87.0** | **57.69** | **75.0** | **59.15** | **51.54** | **73.37** |
| Qwen3-4B | 71.3 | 54.13 | 75.0 | 65.36 | 56.63 | 78.24 |
| + PPO | 87.4 | 58.07 | 85.0 | 72.03 | 59.29 | 83.10 |
| + GRPO | 87.6 | 59.81 | 85.0 | 76.12 | 60.55 | 83.19 |
| + HEPO | 88.2 | 59.51 | 82.5 | 74.31 | 59.48 | 82.37 |
| + TREERPO | 88.6 | 60.27 | 87.5 | 77.06 | 60.96 | 83.56 |
| + TEMPO | **91.0** | **62.32** | **92.5** | **79.18** | **62.51** | **85.49** |

Table 1: Comparison of PPO, GRPO, HEPO, and TEMPO on mathematical and medical reasoning benchmarks using Qwen3-1.7B and Qwen3-4B as base models. MATH and MedQA are considered *in-distribution* (ID) tasks, while GSM-HARD, AMC23, MedMCQA, and MMLU-Medical are treated as *out-of-distribution* (OOD) evaluations.

## 6.2 COMPUTATIONAL EFFICIENCY

Figure 4 reports validation accuracy against wall-clock training time on Qwen3-4B for both MATH and MedQA, with all runs executed on identical hardware ($2\times$NVIDIA H100 80GB). PPO shows the slowest convergence and lowest final accuracy, while GRPO and HEPO provide more stable training but plateau earlier. In contrast, TEMPO demonstrates clear computational advantages: on MATH, TEMPO achieves GRPO's best accuracy about $1.4\times$ faster, and on MedQA it reaches GRPO's peak roughly $1.6\times$ faster. Moreover, TEMPO continues to improve beyond these points, ultimately attaining higher final accuracy. These results show that integrating tree-structured TD corrections improves both credit assignment and training efficiency under realistic hardware budgets.

## 6.3 EFFECT OF GROUP SIZE

An important hyperparameter in GRPO-style methods is the *group size*, i.e., the number of responses sampled per prompt during training. Larger groups provide a more reliable relative baseline, but also increase computational cost. Figure 5 shows the effect of varying group size $(3, 5, 7, 9)$ on MATH accuracy for Qwen3-1.7B. We observe two key trends. First, increasing group size improves performance for both GRPO and TEMPO, consistent with prior findings that larger groups yield stronger learning signals. Second, TEMPO consistently outperforms GRPO across all group sizes, with gains of around $0.5-2$ points in accuracy.

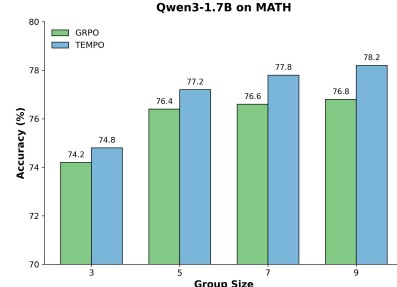

Figure 5: Effect of group size on MATH accuracy for Qwen3-1.7B. TEMPO consistently outperforms GRPO across all settings.

## 6.4 WHERE DO BRANCHES HAPPEN?

Figure 6 visualizes the tokens that most often appear at *branch nodes*—prefixes $s_t$ with multiple continuations or high next-token entropy—aggregated over TEMPO training. On Math, branching tokens cluster around *planning and formalization* phrases such as "let", "find", "solve", "suppose", numerals, variables, and operators (e.g., "$x$", "equation", "series", "triangle"). These are the points where the model chooses a solution strategy (set up variables, recall a fact, pick a formula) before committing to derivations. TEMPO's TD correction therefore acts exactly where the plan can diverge (e.g., choosing the wrong identity vs. the right one). In contrast, medical branching tokens emphasize *clinical entities and constraints*: demographics ("year-old", "man/woman"), symptoms ("pain", "fever", "cough"), disease terms ("diabetes", "seizure"), and linking words that steer differentials ("with", "who", "which"). These tokens define the candidate diagnosis/workup branches, so TEMPO focuses signal where the case interpretation can split. Across domains, branches coincide with high-stakes decision tokens the junctures that determine the downstream trajectory. This qualitative evidence complements our quantitative results: the tree-aware TD signal is delivered exactly where it matters most.

## 6.5 EFFECT OF BRANCH COUNT

To study the effect of branching toward training performance, we build the prefix tree at the first epoch and record the number of branches $k$. Responses prefix trees are then grouped by their initial $k \in \{7, 8, 9, 10, 11\}$, and we track their training accuracy over subsequent epochs. As shown in Figure 7, responses that begin with more branches learn faster and reach higher accuracy, reflecting the availability of more branch tokens where TEMPO applies non-zero TD corrections. When the observed branching is minimal ($k=G=7$), the TD signal largely vanishes and TEMPO behaves like GRPO, yielding the slowest improvement among the others.

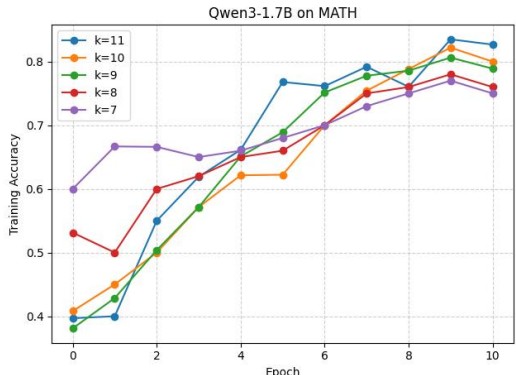

Figure 7: Training accuracy vs. epochs for different numbers of preserved branches $k$ at group size $G=7$.

## 6.6 ABLATION STUDY

We decompose TEMPO into two increments over GRPO: (i) P2T, which computes a nonparametric prefix value by aggregating descendant outcomes in the response-group prefix tree, and (ii) branch-gated TD, which injects token-level TD corrections only at branch tokens. As shown in Table 2, P2T alone yields a clear gain over GRPO (75.2% → 77.0%), indicating that tree-derived prefix values already sharpen credit. Adding the TD term further improves performance (77.0% → **78.4%**), confirming that localized TD at decision points complements the group-normalized signal without introducing a learned value network or extra supervision.

| Method | Accuracy (%) |
|---|---|
| GRPO | 75.2 |
| + P2T | 77.0 |
| + P2T + TD(TEMPO) | **78.4** |

Table 2: Ablation study for Qwen3 1.7B on MATH. Adding the prefix-to-tree baseline (P2T) to GRPO improves accuracy; adding branch-gated TD on top yields the full method (TEMPO) with the best performance.

## 6.7 GENERALIZATION

Table 1 summarizes performance on both in-distribution (ID) and out-of-distribution (OOD) benchmarks for mathematics and medicine. On the ID tasks (MATH and MedQA), TEMPO consistently achieves the highest accuracy, surpassing TREERPO, PPO, GRPO, and HEPO across both Qwen3-1.7B and Qwen3-4B. For example, TEMPO improves MedQA accuracy from 76.1% (GRPO, 4B) to 79.2%, and raises MATH accuracy from 87.6% to 91.0%. On OOD evaluations, TEMPO also establishes clear gains. In the math domain, it pushes GSM-HARD accuracy from 59.8% (GRPO,

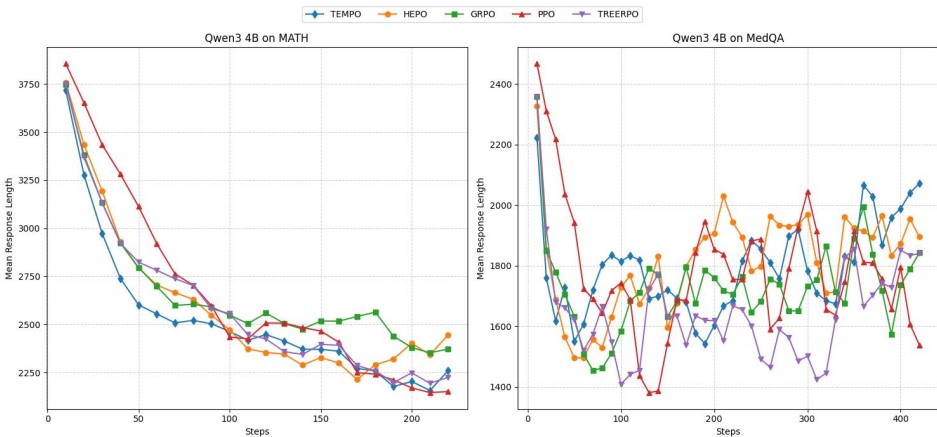

Figure 8: Mean Responses Length of MATH and MedQA for Qwen3- Qwen3-4B accross training.

4B) to 62.3%, and AMC23 from 85.0% to 92.5%. In the medical domain, it improves MedMCQA from 60.55% to 62.51% and MMLU-Medical from 83.2% to 85.5%. These improvements across unseen distributions highlight that TEMPO not only enhances in-distribution learning efficiency but also yields stronger generalization to harder and more diverse reasoning tasks.

# 7 CONCLUSION

In this work, we introduced TEMPO, a reinforcement learning algorithm for LLM alignment that integrates temporal-difference (TD) signals into group-relative optimization by exploiting the tree structure of sampled responses. Unlike PPO, which requires training a separate value network, and GRPO, which discards token-level information by relying purely on Monte Carlo signals, TEMPO unifies the strengths of both approaches without additional model components. By deriving value estimates directly from the response tree, TEMPO enables token-level TD corrections on top of group-relative normalization, yielding more fine-grained and stable credit assignment. Our experiments across mathematics and medicine demonstrate two key findings. First, TEMPO achieves higher accuracy than PPO, GRPO, and HEPO on both in-distribution and out-of-distribution benchmarks, showing strong generalization. Second, TEMPO converges significantly faster in wall-clock time, achieving comparable or better accuracy up to $1.6\times$ earlier under the same hardware configuration. Overall, TEMPO establishes a practical and scalable approach to reinforcement learning with verifiable feedback. It provides fine-grained credit assignment without the overhead of a value model, improves training efficiency, and enhances robustness across domains.

## ETHICS STATEMENT

All authors have read and will adhere to the ICLR Code of Ethics. We acknowledge that the Code applies to all aspects of conference participation (submission, reviewing, discussion).

**Scope and compliance.** This work studies RL for LLM reasoning using public benchmarks only. No human subjects were recruited and no new personal data were collected; therefore, IRB approval was not required.

**Data, privacy, and licensing.** We use publicly available datasets (MATH, MedQA, GSM-HARD, AMC23, MedMCQA, MMLU-Medical) under their licenses. To the best of our knowledge, these datasets do not contain personally identifiable information. Evaluation relies on programmatic, verifiable feedback (e.g., exact match), not human raters or proprietary judges.

**Safety and misuse.** Models are research artifacts and *not* clinical decision tools. Medical benchmarks are knowledge tests; deploying models downstream for healthcare or other high-stakes uses requires additional validation, domain oversight, and regulatory compliance.

**Fairness and limitations.** Benchmarks may encode societal or domain biases. We report results across multiple tasks and discuss limitations (e.g., distribution shift, overfitting risk). We encourage cautious interpretation beyond the evaluated settings.

**Transparency and reproducibility.** We document training settings, hardware, and evaluation protocols; we intend to release code/configs subject to third-party licenses. No hidden reward/process models, teachers, or special samplers were used. For details on LLM usage in paper writing, see Appendix A.8.

**Conflicts of interest.** Any potential conflicts, sponsorships, or competing interests will be disclosed in the author checklist.

## REPRODUCIBILITY STATEMENT

We provide full details to ensure reproducibility. Dataset sources and splits are in Appendix A.2; implementation details and training practices are in Appendix A.3; hyperparameters are listed in Appendix A.4 and Table 3; compute setup is described in Appendix A.5; and the software stack is documented in Appendix A.6. We also include an anonymized code repository link in Appendix A.7.

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

# A APPENDIX

## A.1 GENERALIZATION ANALYSIS

To further investigate the gap, we analyze the relationship between training and validation accuracy on MedQA for Qwen3-4B (Figure 9). We find that PPO exhibits clear overfitting: its training accuracy continues to increase while validation accuracy plateaus, indicating weak generalization. In contrast, both GRPO and TEMPO improve training and validation accuracy in tandem, with TEMPO achieving the highest performance on both, suggesting more reliable generalization rather than memorization of the training distribution. This analysis explains why PPO lags behind in final MedQA accuracy and underscores TEMPO's advantage when scaling to larger models and more challenging domains.

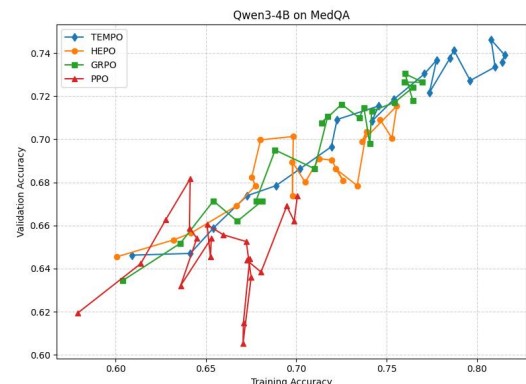

Figure 9: Training vs. validation accuracy on MedQA with Qwen3-4B. PPO overfits to training data, while TEMPO maintains better generalization.

## A.2 DATASETS AND MODELS

We consider two domains: mathematics and medicine. For training, we adopt one representative dataset from each domain: MATH (Hendrycks et al.) for mathematical reasoning and MedQA (Jin et al., 2021) for medical question answering. These serve as the *in-distribution (ID)* training tasks. For evaluation, we test on both the in-distribution test sets of MATH and MedQA, as well as multiple *out-of-distribution (OOD)* benchmarks to assess generalization. In the math domain, OOD benchmarks include GSM-HARD (Gao et al., 2023), a challenging variant of GSM8K with harder grade-school problems, and AMC23[2], a set of recent American Mathematics Competition problems. In the medical domain, OOD benchmarks include MedMCQA (Pal et al., 2022), a dataset con- sisting of multiple-choice medical questions designed to test clinical knowledge, and MMLU-Medical (Singhal et al., 2023), a medical subset of the Massive Multitask Language Understanding (MMLU) benchmark focusing on diverse topics in the medical field. We adopt two publicly available models from the Qwen 3 (Yang et al., 2025a) family: Qwen3-1.7B and Qwen3-4B. Both models are fine-tuned in our experiments under identical settings to ensure fair comparison.

## A.3 IMPLEMENTATION DETAILS

To ensure our GRPO implementation is robust, and our evaluation reflects its full potential, we have applied a set of wellestablished techniques and best practices from the literature Yu et al. (2025). Below, we outline the key implementation details that were most effective in our experiments:

- Clip-Higher (decoupled clipping). We decouple the clipping bounds and raise the upper cap $(1 + \varepsilon_{high})$ while keeping the lower cap $(1 - \varepsilon_{low})$, which allows low-probability "exploration" tokens to increase more freely and helps prevent entropy collapse.

- Token-level policy-gradient loss: Token-level policy-gradient loss (global token averaging): We optimize a token-level surrogate averaged over *all* tokens in the batch, broadcasting each response's group-normalized outcome reward to its tokens since sample-level averaging underweights long responses and fails to penalize low-quality long patterns, which destabilizes training; token-level loss restores balanced credit assignment and yields healthier length/entropy dynamics.

---

[2]https://huggingface.co/datasets/AI-MO/aimo-validation-amc

- Remove KL divergence: In long-CoT reasoning, the online policy can beneficially diverge from the initialization; thus we omit an explicit KL regularizer and rely on clipping for stability.

**Training Details and Hyperparameters** We adopt a binary task reward R that evaluates final answer correctness against ground truth, following previous work Huang et al. (2024); Ivison et al. (2024). To ensure fair comparison, all methods consume the same number of episodes during training: for each question, we sample 6 episodes and go over the dataset 10 times, yielding 60 episodes per question across all methods.

## A.4 HYPERPARAMETER

In this section, we provide a comprehensive overview of the hyperparameters used in our experiments. The number of training episodes was carefully selected to ensure that the amount of training data remained consistent across all methods.

**PPO** Finetuning LLMs with PPO is known to be highly sensitive to hyperparameter choices, making optimal selection critical for strong performance. To ensure robustness, we considered hyperparameter values reported in prior studies Shao et al. (2024) and performed extensive sweeps across a wide range of candidate values. Specifically, we first identified the set of hyperparameters that achieved the best performance across both the MATH and MedQA tasks using the Qwen3 1.7B model. This optimal configuration was then employed for the remainder of our experiments. The complete list of PPO hyperparameters, along with their respective search spaces, is shown in Table 3.

**GRPO, HEPO, and TEMPO** Since policy optimization in RLOO and GRPO closely resembles PPO, we initialized their hyperparameters using the PPO configuration. This ensures a strong starting point while enabling a systematic comparison among the algorithms. We note that the absence of explicit credit assignment in these methods may result in high-variance policy gradient updates, potentially leading to instability Greensmith et al. (2004). The full list of hyperparameters for GRPO, HEPO, and TEMPO is provided in Table 3.

## A.5 COMPUTE

All experiments were conducted using multi-GPU training to efficiently handle the computational demands of large-scale models. For the Qwen3-1.7B model, we utilized a node with 1 × Nvidia H100 80GB GPUs to train both TEMPO and all the baselines. For the larger Qwen3-4B model, we employed a more powerful setup, using a node with 2 × Nvidia H100 80GB GPUs.

## A.6 SOFTWARE STACK

For model implementation, we utilize the Huggingface library. Training is carried out using the VERL Zhang et al. (2024) distributed training library, which offers efficient multi-GPU support. For trajectory sampling during RL training, we rely on the vLLM library Kwon et al. (2023), which provides optimized inference for LLMs.

## A.7 REPRODUCITY

In this study, all experiments were conducted using open-source libraries, publicly available datasets, and open-weight LLMs. To ensure full reproducibility, we will make our codebase publicly available on GitHub at `https://anonymous.4open.science/r/tempo-0AF2`.

## A.8 LLM USAGE

In accordance with the ICLR 2026 policies on LLM usage, we disclose how LLMs were used in this work. LLMs were employed to assist with grammar polishing, wording improvements, and drafting text during paper preparation. All technical content, proofs, experiments, and analyses were conceived, implemented, and validated by the authors. Authors remain fully responsible for the correctness of the claims and results.

| Parameter | Value | Notes |
|---|---|---|
| **Training** | | |
| Optimizer | AdamW | |
| Adam parameters $(\beta_1, \beta_2)$ | (0.9, 0.999) | |
| Learning rate | $1 \times 10^{-6}$ | |
| Weight decay | 0.0 | |
| Warmup | 0% of training steps | |
| # Train steps (MATH) | 220 steps | $\sim$10 dataset epochs |
| # Train steps (MedQA) | 420 steps | $\sim$10 dataset epochs |
| **General** | | |
| Maximum prompt length | 1024 tokens | |
| Maximum response length | 8192 tokens | |
| Training batch size | 512 | |
| **PPO** | | |
| Mini-batch size | 64 | |
| # Inner epochs per PPO step | 2 | |
| Discount factor $\gamma$ | 1.0 | |
| GAE parameter $\lambda$ | 1.0 | |
| KL penalty coefficient $\beta$ | $1 \times 10^{-4}$ | |
| **GRPO/HEPO/TEMPO** | | |
| # Responses per prompt | 6 | |
| Mini-batch size | 64 | |
| Discount factor $\gamma$ | 1.0 | |
| KL penalty coefficient $\beta$ | 0.0 | |
| Policy clipping parameter $\epsilon$ | 0.28, 0.2 | |
| **HEPO** | | |
| $\rho$ | 0.2 | Only do gradient update on top 20% high entropy tokens |

Table 3: Summary of hyperparameters used in the experiments.

No LLMs were used to generate research ideas, write code for experiments, or produce results. No confidential information was shared with LLMs, and no prompt injections or other inappropriate uses were involved.

This disclosure aligns with the ICLR Code of Ethics: contributions of tools are acknowledged, while accountability and verification rest entirely with the human authors.

