# OpenReview forum: "EXPLOITING TREE STRUCTURE FOR CREDIT ASSIGNMENT IN RL TRAINING OF LLMS"
_ICLR.cc/2026/Conference — ICLR 2026 Conference Withdrawn Submission_

### Official Review · Reviewer_BAWo · 2025-10-17

**Soundness:** 3
**Presentation:** 3
**Contribution:** 3
**Rating:** 6
**Confidence:** 5

**Summary:**

This paper tackles the challenge of reward sparsity in RLVR. The authors introduce Prefix-to-Tree (P2T), a procedure that aggregates a group of responses into a prefix tree to compute a value function based on descendant outcomes. Building on P2T, they propose TEMPO, an algorithm designed to restore token-level credit assignment and mitigate reward sparsity in GRPO. TEMPO is evaluated on both in-distribution and out-of-distribution (OOD) benchmarks, demonstrating its robustness.

**Strengths:**

- The paper is clearly written and easy to follow.

- The proposed method is straightforward to implement, improving performance while maintaining the same training cost as the original GRPO framework.

- The approach is intuitively sound and memory-efficient, as it eliminates the need to train an additional value model to provide temporal-difference signals.

- TEMPO demonstrates robustness with small sample sizes. Although a potential concern is that TEMPO might require large group sizes to be effective, the experimental results show that it consistently outperforms the GRPO baseline even with small group sizes ( $\leq$ 9).
Weaknesses

**Weaknesses:**

- **Concern Regarding Novelty**

While the proposed method is appealing and intuitively sound, its novelty may be limited. As the authors acknowledge in Section 2, prior work [1] has also introduced a tree-search procedure into the GRPO framework. The similarity between TEMPO and TreeRPO raises a concern about the distinct contribution of this work.

[1] TreeRPO: Tree Relative Policy Optimization, arxiv 2025.

- **Missing Baseline**

TreeRPO [1] shares a great similarity with the proposed method TEMPO, yet it is currently absent from the main experimental results.

- **Lack of Justification for Advantage Computation**

The justification for the advantage calculation at line 299 requires clarification. Specifically, the rationale for standardizing the TD error by dividing it by $\text{std}(r)$ is unclear. A more conventional approach would be to normalize by the standard deviation of the TD error itself (e.g., $\text{std}(V(s_{t+1})-V(s_t))$). The authors should explain the motivation behind their chosen normalization method.

**Questions:**

See about.

---

> ### Author Response · Authors · 2025-11-18
> **Response to Reviewer BAWo**
>
> Thanks for the careful read and concrete pointers. We address each point and have updated the draft accordingly.
>
> 1) Novelty vs. TreeRPO
> We agree TreeRPO is a closely related, tree-based line; our contribution targets a different point on the accuracy–efficiency frontier:
>
> Implicit vs. explicit trees. TEMPO builds the implicit prefix tree from the standard GRPO group of rollouts—no new sampler, no step-wise reward/judge. TREERPO performs explicit tree sampling/search to densify step signals.
>
> Critic-free, branch-gated TD. TEMPO computes a nonparametric prefix value V(s_t) from descendants and adds TD only at branch tokens; non-branch tokens revert exactly to GRPO (stable, zero-TD).
>
> Same loop/cost as GRPO. TEMPO keeps GRPO’s training loop and sampling budget; explicit tree search typically incurs higher rollout cost.
>
> 2) Missing TREERPO baseline
> We have added TreeRPO to the main results. Under matched hardware and sampling budgets, TEMPO attains higher validation accuracy with less wall-clock time. (We note the TreeRPO paper’s GitHub link appears to point to a different project; we therefore re-implemented TREERPO from the paper to the best of our understanding and will release our reproduction and configs. Results may not identically match the authors’ private code.)
>
> 3) Justification for the advantage normalization (line 299)
> Our advantage has two terms: a group-normalized outcome signal (GRPO’s Monte-Carlo, response-level) and a branch-gated TD correction (token-level). We standardize the MC term in the usual GRPO way using the group mean/std. For the TD term, we scale by the same group std. The motivation is simple:
>
> Balance of signals. We want equal a priori importance of response-level (MC) and token-level (TD) signals on a comparable numeric scale. Using the same group std for both avoids the TD term dominating (or vanishing) when its own sample variance is small/unstable at early training or when branching is sparse.
>
> We appreciate the thoughtful feedbacks and we welcome any further comments/suggestions.

---

### Official Review · Reviewer_HxN3 · 2025-10-20

**Soundness:** 2
**Presentation:** 2
**Contribution:** 2
**Rating:** 2
**Confidence:** 4

**Summary:**

This paper presents TEMPO, an RLHF training paradigm that introduces branching within rollouts to address the challenge of sparse rewards. Specifically, TEMPO generates multiple branches from each rollout, groups their final rewards to derive intermediate rewards, and integrates temporal-difference signals for improved training efficiency and stability. Evaluation results show encouraging speedup and better quality.

**Strengths:**

+ Improving RLHF efficiency and quality is an important problem.
+ The paper is overall well-written.
+ The design is straightforward and reasonable.
+ Evaluations report encouraging improvements.

**Weaknesses:**

- The contribution appears incremental, particularly compared to existing tree-based methods such as TreeRPO.
- The evaluation is limited, lacking comparisons with highly relevant baselines like TreeRPO.

**Questions:**

Thank you for submitting your work to ICLR. The paper tackles an important problem, and the idea of branching to collect finer-grained rewards is reasonable. However, the proposed design seems conceptually close to existing tree-based RLHF approaches, and the paper's discussion on novelty over prior work (e.g., TreeRPO) is not entirely convincing. The absence of direct comparison experiments further weakens the claim of contribution.

- Q1: It is unclear how and when the rollout is split to create new branches. Could the authors clarify the branching criterion and timing?

- Q2: The paper uses a fixed number of branches per prompt. It would be better to adapt it.

- Q3: The paper should consider adding comparison experiments against other tree-based approaches, such as TreeRPO, to strengthen the evaluation.

- Q4: While both branching and temporal-difference learning appear beneficial, an ablation study would help isolate the contribution of each component.

---

> ### Author Response · Authors · 2025-11-18
> **Response to Reviewer HxN3**
>
> Thank you for the thoughtful review and for emphasizing comparisons to tree-based methods. We’ve revised the paper to address novelty, evaluation breadth, and clarity.
>
> On novelty vs. TREERPO / “incremental” framing.
> Our method differs conceptually and operationally from TREERPO and related tree-sampling approaches:
>
> No explicit tree sampler. We use the implicit prefix tree naturally induced by the G responses per prompt (standard GRPO sampling). We do not branch via a new sampler nor require process rewards or step-wise teachers.
>
> Nonparametric prefix value + branch-gated TD. We compute a critic-free, nonparametric prefix value from all descendants sharing then add a TD correction only at branch tokens; at non-branch tokens the TD term is zero and the update reduces to GRPO.
>
> GRPO-compatible, same loop/cost. Unlike tree-sampling methods that expand trajectories to densify rewards, our updates stay in the GRPO loop with the same sampling budget and lower wall-clock.
> These choices target a different point on the accuracy–efficiency frontier: tree-aware, token-level credit without extra sampling or learned critics.
>
> On evaluation breadth (TREERPO).
> We have added TREERPO as a baseline. Under matched hardware and sampling budgets, our method attains higher validation accuracy with less wall-clock (new Fig. 4). This strengthens the claim that implicit-tree TD credit is a competitive (and efficient) alternative to explicit tree sampling.
>
> Reproducibility note. The TREERPO paper links to a GitHub repo that appears to correspond to a different project; we could not locate an official TREERPO implementation. We therefore re-implemented TREERPO from the paper to the best of our understanding and verified settings carefully. While we aimed for fidelity, our implementation may not reflect the authors’ system exactly. We will open-source our reimplementation and configs for transparency.
>
> Responses to questions
>
> Q1. How/when is branching created?
> We do not force splits. Given G rollouts per prompt, we build the implicit prefix tree by aligning token positions across the G responses. We compute the nonparametric V(s_t)
>  from the subtree and apply TD only at these branch tokens; else where the TD term is zero (falls back to GRPO). This happens every training step for each prompt’s group.
>
> Q2. “Fixed number of branches per prompt—should be adaptive.”
> We do not fix the number of branches; it is induced by the data (G, temperature/top-p, task). In ablations we stratify by observed branch count to study learning dynamics (not to fix it). Practically, adaptivity comes from the sampling distribution: higher temperature or larger G yields more branching—and thus more TD opportunities—without changing the algorithm.
>
> Q3. Add comparisons against other tree-based approaches (TREERPO).
> Done.
>
> We now include TREERPO using authors’ settings where possible. Our approach outperforms TREERPO with lower wall-clock under matched budgets (Fig. 4), indicating that dense step-signals via explicit branching are not required to realize strong gains.
>
> Q4. Provide ablations to isolate contributions.
> Added.
>
> We include an ablation study in Appendix Table 2.
>
> We hope those clarifications and revisions addressed the reviewer's concern. Please let us know if there's any further comments.

---

> > ### Comment · Reviewer_HxN3 · 2025-11-27
> >
> > Thank you for the response; some of my concerns have been addressed. I have raised my score, though I still have reservations about the incremental nature of the contribution.

---

> > > ### Author Response · Authors · 2025-11-29
> > > **Response to Reviewer HxN3**
> > >
> > > Thank you for the reconsideration and for raising your score. On the “incremental” concern: TreeRPO and other tree-based papers appeared on arXiv only 1-2 months before our ICLR submission, i.e., essentially contemporaneous with our work. We developed TEMPO independently, and we’ve clarified the distinctions:
> > >
> > > Mechanism: TEMPO uses the implicit prefix tree from standard GRPO sampling, computes a nonparametric prefix value 𝑉(s_t)​, and applies branch-gated TD only at decision tokens—reverting exactly to GRPO elsewhere.
> > >
> > > Cost/loop: TEMPO keeps the GRPO loop and wall-clock budget, whereas explicit tree sampling generally increases rollout cost.
> > >
> > > Evidence: We included TreeRPO as a baseline; under matched hardware/sampling, TEMPO converges faster and reaches higher validation accuracy. Ablations (GRPO → +P2T → +P2T+TD) isolate each contribution.
> > >
> > > Given the timing (contemporary arXiv releases) and the substantive design/efficiency differences plus new comparisons and ablations, we believe TEMPO represents a distinct and practical contribution. We appreciate your updated assessment and are happy to incorporate any further suggestions.

---

### Official Review · Reviewer_hgBX · 2025-11-01

**Soundness:** 4
**Presentation:** 4
**Contribution:** 2
**Rating:** 2
**Confidence:** 4

**Summary:**

This paper claims that by grouping the shared prefixes among a group of responses to a query when doing GRPO, one can extract fine-grained credit assignment. They show that this leads to gains in mathematical and medical tasks claiming this proves its generality. They analyze the branching tokens and the effect of the size of the group on the results providing further intuition.

**Strengths:**

I think the paper correctly claims that credit assignment is troublesome in current reasoning models. Exploiting the shared prefixes, if they exists, is a smart way to extract fine-grained credit assignment. The paper really does go deep in analyzing this behaviour and provides insights. It does a thorough related work section.

**Weaknesses:**

The problem with the work is that, it does not lead to significantly better credit assignment. The main problem is, the responses from a diverse reasoning model quickly diverge from each other and therefore, aside from the maybe first few thousand tokens, the rest will only have one response. It is very intuitive that this happens: the paths simply diverge due to sampling of tokens. This is very limiting. I have confidence that what authors have shown really holds: the extraction of the extra juice left in the shared prefixes for more accurate credit assignment is useful. But, the problem is that, the effects stop there. It cannot substantially improve: there is no knob to extract finer credit assignment. This paper can be a nice addition to a GRPO run, but it is not a significant contribution: it has small impact.

Aside from the above, I am wondering why the baseline numbers reported for MATH500 are different from Qwen3 official report. In theirs, the Qwen3-4B performance is 97% on MATH500 in thinking mode. However, the authors train and it starts from 82 and goes to 87% on validation. This seems contradictory to me and indicates maybe the authors tested their method on non-thinking mode. I think this is the case especially as the authors set the maximum response length to 1024 tokens: this is very low for reasoning models. I think if the authors tested their method on the correct setup, with high number of max responses tokens and in thinking mode, their method would have significant trouble because aside from the the first thousand tokens, the rest will only have just one response due to diverging branches.

Unfortunately, these two weaknesses leaves me with no choice but to reject the paper.  I think the results of the paper hold and are correct, but does not scale to actual training runs with long reasoning chains.

**Questions:**

1-Are you using the thinking mode of the said models or their non-thinking mode?
2-If you set the maximum response length to a standard number, preferably 32K, if compute is limited 8K, would the results hold?

---

> ### Author Response · Authors · 2025-11-18
> **Response to reviewer hgBX**
>
> Thank you for your thoughtful review. We address each point and clarify our setup below; we’ve also updated the draft accordingly.
>
> 1: “Implicit branching gives only small gains; paths quickly diverge; there’s no knob for finer credit.”
> Our premise is that most outcome-moving decisions concentrate at early/mid “branch tokens,” not uniformly across the whole sequence. This matches our diagnostics: Where the TD fires. Branch tokens (high-entropy forks) localize at strategy/diagnosis pivots; non-branch tokens receive zero TD and we safely fall back to GRPO (no harm, no noise injection).
>
> Does implicit branching suffice?
> Yes. We added TREERPO (explicit tree sampling with denser step signals) as a baseline and TEMPO still outperforms it in accuracy with lower wall-clock time, because explicit tree sampling demands substantially more rollouts. This supports our hypothesis: precise credit at the few key tokens is sufficient; dense credit everywhere is not required in practice.
>
> “No knob” concern. While we don’t require a special sampler, there are controllable levers that increase shared prefixes (and thus TD opportunities): group size G, (in our ablations) the observed branch budget correlates with faster learning. Our ablation study (Table 2) shows incremental, robust gains. In short, TEMPO is more than a “nice add-on”: it offers targeted token-level TD with GRPO-like simplicity and cost, and scales better than explicit tree-sampling methods under matched compute.
>
> 2: “1k max length; long-horizon chains not exercised; maybe non-thinking mode.”
> We apologize for the confusion. 1k refers to the prompt max length, not the response. Our max response length is 8192; we corrected Table 3 in the revision and added Fig. 8 (mean response length over training) to show 8k comfortably covers our trajectories. We also use the model’s thinking mode, not the no-think mode.
>
> Clarifications on datasets:
>
> MATH vs. MATH500: Our math experiments use the MATH benchmark of Hendrycks et al, not MATH500. MATH has both training and test subset and that's the main reason why we choose to experiment with. The 97% figure cited for Qwen3-4B is for MATH500 in “thinking” mode, a different dataset. Our numbers aren’t contradictory; they target MATH under RL training, not one-shot eval on MATH500.
>
> Measuring mathematical problem solving with the math dataset(Dan Hendrycks, Collin Burns, Saurav Kadavath, Akul Arora, Steven Basart, Eric Tang, Dawn Song, and Jacob Steinhardt)
>
> Would results hold with 32k (or 8k if limited)?: We already train with 8k responses; Fig. 8 shows it is sufficient for the chains we generate.
>
> We hope these updates address the concerns and we welcome any further suggestions.

---

### Official Review · Reviewer_kGDS · 2025-11-01

**Soundness:** 2
**Presentation:** 3
**Contribution:** 2
**Rating:** 2
**Confidence:** 4

**Summary:**

This paper tackles the token-level credit assignment challenge in reinforcement learning for large language models with verifiable rewards (e.g., math or medical reasoning). The authors propose TEMPO. TEMPO is a critic-free method that constructs a prefix tree from multiple sampled responses per prompt, computes nonparametric prefix values, and introduces branch-aware corrections. TEMPO provides fine-grained credit at branching tokens while preserving the efficiency and simplicity of GRPO-style training. Empirical results on MATH, MedQA, and several out-of-distribution benchmarks show that TEMPO outperforms PPO, GRPO, and HEPO in both convergence speed and accuracy using Qwen3-1.7B/4B models.

**Strengths:**

1. The studied problem is important. Sparse reward is a key limitation of current RL for LLMs, which can cause sub-optimal optimization.

2. The proposed idea is clear and easy to understand. Using a tree to calculate a more accurate reward for each token can be a good way to introduce such dense rewards.

3. The writing is clear and easy to understand.

**Weaknesses:**

1. The proposed method does not explicitly control the branch of the repeated sampling tree, but relies on implicit branching across sampled responses. In many prompts, only a small portion of tokens can actually form branch points, so most updates may reduce to GRPO with sparse rewards. Some further analysis can be conducted to verify why implicitly building the repeated sampling tree is a good choice and introduces enough dense reward signals.

2. Another concern is related to the experiment setting. As illustrated in Table 2, the maximum response length is only around 1k tokens, which is far shorter than the original model’s typical response length (up to 40k). Such a limited sequence length makes it difficult to fully justify the training effectiveness of the proposed method, as long-horizon reasoning and delayed credit assignment are precisely where token-level advantages should matter most. Moreover, this limitation is closely tied to the previous point on implicit branching; when the response length is long, the later tokens may not share any branches with other responses, which makes the reward very sparse.

3. The training step is also limited. Usually, in recent RL papers, > 800 training iterations are needed to make sure training converges. However, the number of training iterations on Math is 220, which can be too few. Despite that I feel that the overall idea is neat, but the evaluation part is not enough to support the major hypothesis of the paper.

**Questions:**

See weakness.

---

> ### Author Response · Authors · 2025-11-18
> **Response to reviewer kGDS**
>
> Thanks for the thoughtful and constructive feedback. We address each point below and have updated the draft accordingly.
>
> 1: Implicit branching vs. dense signals.
> Our method intentionally uses the implicit prefix tree induced by repeated sampling, avoiding extra samplers and retaining GRPO’s drop-in loop. We now include TREERPO—which performs explicit tree sampling to densify credit—as a baseline. Under identical hardware, our method still converges faster and to higher accuracy (see Fig. 4), while explicit tree sampling incurs additional rollout overhead. Empirically, the TD term in our advantages fires exactly where decisions occur: branching tokens concentrate at strategy/diagnosis pivots (Fig. 6), and higher initial branch counts predict faster learning and higher final accuracy (Fig. 7). An ablation shows that (i) prefix-to-tree values (P2T) already help over GRPO and (ii) adding branch-gated TD yields the full gain (Table 2). Together, these analyses support that implicit trees provide sufficiently informative (and well-targeted) token-level signals without the sampling cost of explicit tree construction.
>
> 2: Sequence length clarification.
> We apologize for the confusion. The 1k limit is the maximum prompt length, not the response length. Our experiments use 8192 as the maximum response length, which we have clarified in the revision Table 3. Figure 8 reports mean response length across training on both domains, showing that 8k comfortably covers our trajectories in practice; thus the long-horizon, delayed-credit regime is exercised under our settings.
>
> 3: Training iterations and convergence.
> To make the training more stable, we train with a large effective batch (512 samples), which reduces the number of outer iterations while maintaining total episode budget. We added the information to Table 3 of the revision draft. Validation curves indicate clear convergence under our schedule; adding more iterations leads to overfitting (training reward increase, validation reward stagnant or decrease), as shown in Figure 9.
>
> We appreciate the reviewer’s suggestions. We hope these updates address the concerns; we’re happy to incorporate any further suggestions.

---

### Note · Authors · 2026-01-06

I have read and agree with the venue's withdrawal policy on behalf of myself and my co-authors.